# Rodent Models of Diabetic Retinopathy as a Useful Research Tool to Study Neurovascular Cross-Talk

**DOI:** 10.3390/biology12020262

**Published:** 2023-02-07

**Authors:** Karolina Polewik, Maciej Kosek, Daniel Jamrozik, Iwona Matuszek, Adrian Smędowski, Joanna Lewin-Kowalik, Marita Pietrucha-Dutczak

**Affiliations:** 1Department of Physiology, Faculty of Medical Sciences in Katowice, Medical University of Silesia, Medyków 18, 40-752 Katowice, Poland; 2GlaucoTech Co., Gen. Władysława Sikorskiego 45/177, 40-282 Katowice, Poland

**Keywords:** diabetic retinopathy, diabetes mellitus, retinal diseases, visual impairment, animal models, streptozotocin, alloxan

## Abstract

**Simple Summary:**

Diabetic retinopathy is a progressive complication of diabetes leading to vision loss. About one in three diabetic patients has some degree of diabetic retinopathy. These ocular complications cause emotional distress, reduce the person’s quality of life, and generate huge costs associated with the treatment. The costs are not limited to the provision of ophthalmological care but also relate to psychological or psychiatric care. Understanding the mechanisms leading to the development of diabetic retinopathy is crucial to delay the progression of the disease. Animal models are helpful tools for studying the pathogenesis of diabetic retinopathy and for testing the efficiency of new therapeutic strategies. This review focuses on the rodent models commonly applied in laboratory practice and, importantly, includes suggestions for their application and explains the disadvantages of each model. In addition, the overview is enriched with the current protocols dedicated to the main chemically induced models used in research in the last three years, together with detailed descriptions of the amounts and the frequencies of administered doses of these chemical substances. We hope that this compendium will be useful in selecting the appropriate model for the analyzed aspect of diabetic retinopathy.

**Abstract:**

Diabetes is a group of metabolic diseases leading to dysfunction of various organs, including ocular complications such as diabetic retinopathy (DR). Nowadays, DR treatments involve invasive options and are applied at the sight-threatening stages of DR. It is important to investigate noninvasive or pharmacological methods enabling the disease to be controlled at the early stage or to prevent ocular complications. Animal models are useful in DR laboratory practice, and this review is dedicated to them. The first part describes the characteristics of the most commonly used genetic rodent models in DR research. The second part focuses on the main chemically induced models. The authors pay particular attention to the streptozotocin model. Moreover, this section is enriched with practical aspects and contains the current protocols used in research in the last three years. Both parts include suggestions on which aspect of DR can be tested using a given model and the disadvantages of each model. Although animal models show huge variability, they are still an important and irreplaceable research tool. Note that the choice of a research model should be thoroughly considered and dependent on the aspect of the disease to be analyzed.

## 1. Introduction

### 1.1. Diabetic Retinopathy—The Ocular Complication of Diabetes Mellitus

Diabetic retinopathy (DR), one of the most common chronic complications of diabetes mellitus (DM), is a major cause of visual impairment in adults in developed countries [1]. This disease affects patients suffering from both type 1 diabetes (T1D) and type 2 diabetes (T2D), whereas a considerably higher percentage of patients with T1D develop proliferative retinopathy, which is burdened with a higher risk of extreme visual deterioration [2]. The International Diabetes Federation estimates that the number of diabetic patients may increase to 700 million cases by 2045, and this may consequently lead to a great number of patients with very serious disorders in the visual system (http://www.idf.org, accessed on 31 January 2023). According to statistics, in 2017, the visual dysfunction caused by diabetes affected 2.6 million individuals around the world, accounting for 2.6% of cases of vision loss on the global scale [3].

The considerable diversity of the clinical symptoms of DR hampers the creation of a uniform classification system for this dysfunction. This is not one disease but an entire complex of pathological retinal changes characterized by functional and morphological anomalies of capillary vessels and neuronal cells, typical of diabetes [4]. Consequently, this leads to vision disorders and, frequently, even to vision loss. Unfortunately, in many cases, DR is diagnosed late, and the period from occurrence to diagnosis lasts from seven to ten years on average [5].

As a result of the considerable diversity of the clinical picture of DR, creating a uniform classification system is difficult. Clinical research uses the Early Treatment Diabetic Retinopathy Study classification, but due to its degree of complexity, it is extremely difficult to implement in outpatient practice [6]. The international literature includes a five-degree scale that clearly represents the classification of the diagnosis of the degree of retinopathy [7]. It is presented in Table 1.

### 1.2. The Pathophysiology of the Neurovascular Unit (NVU) in DR

The retinal neurovascular unit is a basic functional unit in the retina and is crucial for the proper visual perception. Different ocular diseases and dysfunction lead to impairment of communication between various retinal cell types forming the NVU structure. NVU includes retinal neurons (i.e., ganglion cells, amacrine cells, horizontal and bipolar cells), glia (Müller cells and astrocytes), and vascular cells (endothelial cells and pericytes), and the interactions between these cells retain environmental homeostasis, ensuring the proper function of the retina. For many years, DR was regarded mainly as a microvascular degeneration characterized by alterations in the integrity of retinal capillaries and their occlusion, vascular leakage, subsequent neovascularization, and retinal haemorrhages. As it advances, compromised retinal microvascular circulation causes ischemia, which increases the size and number of intraretinal haemorrhages. Vascular changes are accompanied by the loss of pericytes and endothelial cells. Pericytes regulate the local capillary blood flow through the expression of α-smooth muscle actin and endothelin (ET-1) receptors and therefore may actively participate in vasoconstriction [8,9,10,11]. ET-1 is a strong vasoconstrictor secreted by retinal endothelial cells [12]. Endothelial cell growth and proliferation are also regulated by pericytes. Pericyte–endothelial interactions are disturbed in many ocular dysfunctions, including DR. It should be noted that vascular cells—pericytes and endothelial cells—are only part of the complicated structure of NVU, and the interactions between the other cells that make up the unit are also important and essential for understanding the pathogenesis of DR. Currently, scientists have defined DR as a neurovascular complication involving progressive disruption of the interdependence between multiple cell types in the retina [6,13]. Moreover, some studies have found evidence of retinal neurodegeneration occurring before the onset of vascular alterations [14,15,16,17,18]. The retinal ganglion cells (RGC) and amacrine cells are the first neurons in which apoptosis is detected. There is currently a growing body of evidence indicating that the damage of RGC appears before the vascular changes and clinical signs of DR. Moreover, RGC apoptosis is preceded by synaptic neurodegeneration and dendritic retraction of these cells [15,17,18,19]. Apoptosis of neuronal cells causes structural and functional alterations in the retina, including the reduction of the retinal layers’ thickness (inner and nerve fibre layers) and the reduction of the amplitude waves in electroretinogram [14,16,20]. Various factors such as glutamate accumulation and toxicity, reduced expression of neurotrophic factors, signalling pathway impairment, and increased production of proinflammatory cytokines (e.g., IL-1β, TNF-α, IL-6) may lead to apoptosis of neuronal cells [19]. Furthermore, parallel to neuronal apoptosis, gliosis of Müller cells and astrocytes occurs, shown by upregulation of glial fibrillary protein (GFAP) [14,21]. Both glial cells are wrapped around retinal blood vessels and are separated from endothelial cells only by a basal lamina. Müller cells, as the functional and metabolic support for neurons, play a critical role in acid–base, ion and water homeostasis, neurotransmitter recycling, and the structural stabilization of the retina, and they modulate immune and inflammatory responses [22,23]. These cells directly and indirectly modify neuronal activity and are an essential structural component of NVU. Whereas, astrocytes are located in the retinal nerve fibre layer (RNFL) and play the significant role in the recruitment and patterning of retina vasculature [24,25]. Moreover, astrocytes initiate the expression of the endothelial cells’ tight junction proteins (zonula occludens-1, ZO-1), whose function and morphology are altered in DR. Endothelial cells of the retina have the highest number of tight junction strands, with the highest complexity, and are frequently entangled with adherens junctions and gap junctions [26]. Several studies have reported that the expression of tight junction proteins is regulated by connexin 43 (Cx43), the gap junction protein, which is decreased in DR [27,28]. The gap junctions are found primarily in astrocytes but also between adjacent endothelial cells and between endothelial cells and pericytes [26]. It needs to be highlighted that many components of tight junction and adherens junction interact with actin and, after releasing from junctions, can translocate to the nucleus and modulate the transcription [29].

In summary, the disruption of interactions between structural components of NVU can play a key role in the development of DR. Therefore, when making an attempt to investigate the mechanisms leading to DR, we should study NVU as a whole rather than analysing the changes taking place in individual cells. Animal models of DR are ideal for carrying out this type of comprehensive research.

## 2. Animal Models Used in DR Research

Animal models are undoubtedly a useful tool for understanding the mechanisms leading to the development of DR, enabling the testing of different treatment methods. A good animal model that maps the physiological or pathological state allows for a relatively transparent analysis of changes taking place during DR. Currently, researchers have mainly focused on rodent models, as shown in Table 2. Rodents have many similarities to humans, especially at the metabolic level. However, it should be emphasized that not all ocular pathologies associated with diabetes can be studied using rodent models. Rodents do not possess a macula; therefore, these models may not be useful in understanding the diabetic macula edema, a common complication of DR [30]. Choosing rodents as models is also justified by the comparatively low costs of breeding, ease of reproduction, and the availability of different research techniques, procedures, and results applied to these animals. Some animal models spontaneously develop diabetes, while others have been genetically manipulated or selectively bred. Chemical inducers or dietary modifications have also been used to induce diabetes.

The next part of this review describes the main characteristics of the most commonly used genetic rodent models in DR research. The subsequent section focuses on the main chemically induced models, with particular reference to the streptozotocin (STZ) model, which we used in our research. Moreover, this section is enriched with practical aspects and contains the current protocols used in research in the last three years. Both parts include suggestions on which aspect of DR can be tested using a given model and the disadvantages of each model.

### 2.1. Selected Genetic Models

Thus far, it has not been possible to identify particular genes that can be held accountable for diabetes, but the essential meanings of some of the genes for the occurrence of this disease have been demonstrated. The changes in the surface molecules of major histocompatibility complex (MHC) class I should be mentioned. They are part of the immune system and are responsible, among others, for the presentation of proteins from intruders entering cells. In patients with diabetes, MHC complexes stimulate lymphocytes to destroy uninfected cells; this is observed in the clinical picture of T1D [31]. Studies have also shown that the presence of HLA-DR3-DQ2 and/or HLA-DR4-DQ8 haplotypes is a significant risk factor for the autoimmunity of pancreatic β-cells, but a trigger factor from the environment is necessary [54].

The topic of the epigenetic modification of gene expression has appeared more often in research into the methods of diabetes diagnosis and treatment. It is a process occurring throughout the entire lifespan of an organism—from conception until death—and is affected by both intracellular and extracellular factors.

#### 2.1.1. Rat Models

Type 1 diabetes

BioBreeding (BB) rats

BB rats are derived through sufficient long inbreeding. BB rats were bred from the Wistar strain as a result of a spontaneous mutation discovered in 1974 in a Canadian colony [31]. They are a model of T1D that shows many similarities of disease symptoms to those appearing in people. On this basis, the genetic, immunological, and environmental factors of the disease can be studied, and new methods of therapy can be developed. MHC genes are responsible for the development of the disease in this type of rat [31,55]. The rat MHC protein complex, designed as an RT1 complex, has two class I loci (A and C) and two class II loci (B and D). The most important region for susceptibility to T1D is the RT1 B/D region, marked as iddm1 [55]. Research on rats of different strains has shown that at least one RT1 class II allele is required for the disease to occur [56,57].

In BB diabetes-prone rats (BBDP), T1D develops spontaneously at 7–12 weeks of age. Diabetes-resistant BB rats (BBDR), despite their genetic similarity to BBDP, do not develop the disease without a suitable trigger factor. The trigger factor can be the regulatory T lymphocyte depletion therapy using the ligation of a toll-like receptor or a viral infection. After induction, full symptomatic diabetes develops rapidly in 2–3 weeks, allowing for the study of all stages of the disease. This model is invaluable for research into the role of environmental factors in the development of diabetes, especially the viral vector [58].

The phenotype of BB rats is characterized by hyperglycemia, hyperinsulinemia, weight loss, and ketonuria requiring insulin treatment. The clinical and metabolic symptoms are preceded by histological abnormalities in the pancreatic islets. The earliest detectable anomaly is islet infiltration by macrophages, starting from the islet periphery through the center and finally covering the whole islet. Macrophages are the initiators of the immune process in the pancreas, and this phase is followed by the migration of CD8+T cells and natural killer (NK) cells, leading to the beta cell destruction process [57].

A major difference between this animal model and human disease is lymphopenia, a decrease in the number of T lymphocytes in the blood, which occurs in BB rats [58].

In BB diabetic rats, after eight weeks, basement membrane thickening of the retinal capillary and the reduction of direct contacts between pericytes and endothelial cells through an increase in the number of caveolae in these cells were observed [59]. Greenwood performed a detailed morphometric analysis of retinal capillary basement membrane thickening at six months and one year post onset of hyperglycemia [60]. After 8–11 months of diabetes, pericyte ghosts and a decrease in the pericyte/endothelial cell ratio were observed [61,62]. Moreover, decreased anionic sites in the luminal plasma membrane of the retinal capillary endothelium and increased pinocytotic transport in the retinal microvasculature of BB diabetic rats were detected [63]. This model was used on DR mainly in the 1980s and 1990s, and it is currently being used to study other changes associated with diabetes mellitus.

Suggested application: to study the response to environmental agents; genetic aspects of diabetes.

Disadvantages: BBDR rats require the application of a trigger factor; lymphopenia (mainly in BBDP); several inbred and outbred lines (BB/Wor, BB/E, BB/Ph)—good selection necessary.

Type 2 diabetes

Zucker diabetic fatty (ZDF) rats

The ZDF model was developed in 1961 [33]. It is mainly characterised by a mutation within the receptor for a signalling peptide called leptin (i.e., the “satiety hormone”), which causes hyperphagia and obesity at around four weeks of age [64]. At 8–12 weeks of age, ZDF rats develop hyperinsulinemia and hyperlipidaemia and show resistance to glucose, making them a good model for T2D [33,34].

Very high levels of cholesterol and triglycerides lead to serious complications typical of obesity: insulin resistance, cardiologic complications, cell dysfunction, and lipoapoptosis, which is programmed cell death due to lipid metabolism problems [65,66].

This animal model is useful to understand the pathogenesis of DR and to test the different therapeutic strategies [67,68,69]. ZDF rats have been used for many years, mainly to study vascular changes in DR [70,71,72,73,74]. Only in recent years have these animals been subjected to intensive analysis of the neuronal changes taking place in DM [69,75,76]. At eight months of age, the animals developed microvascular complications, including an increased number of retinal capillaries and a decreased number of retinal pericytes. Moreover, inflammatory genes, such as TNFα, interleukin-1β, and inducible nitric oxide synthase, were the least abundantly expressed genes, in contrast to VEGF, macrophage migration inhibitory factor, and hypoxia-inducible factor-1 alpha (HIF-1α) genes. Growth factors, such as bFGF and placenta growth factor, and adhesion molecules, such as ICAM-1, were upregulated in diabetic ZDF rats [73]. At 24 weeks of age, ZDF rats showed retinal morphologic degenerative changes, increased neuroretinal thickness, and a decreased number of nuclei in the nuclear layers. Increased apoptosis, particularly in the outer nuclear layer (ONL), inner nuclear layer (INL), and ganglion cell layer (GCL), and the overexpression of glial fibrillary acidic protein (GFAP) with an increased vimentin expression were also observed. A reduction in glutamate–aspartate transporter expression was also detected [75]. Szabo et al. performed a detailed immunocytochemical analysis of the histopathology of both glial and neural cell types in ZDF rat retinas. They reported that the retinas of ZDF rats at the age of 32 weeks were characterised by glia activation, outer segment degeneration for cones and rods, changes in the cone opsin expression patterns, a decreased expression of RPE-65 in pigment epithelial cells, and changes in the labelling patterns and morphology of some amacrine cell types [76]. Warchal et al. analysed for one year the functional changes in the retina and the oxidative state and reported that the amplitude of a-waves and b-waves of ZDF rats was lower than that of healthy (lean) animals and that the expressions of HSP70 and NF*κ*B were higher in diabetes rats. Furthermore, about 60% of the diabetic animals died during the experiment [69].

Suggested application: to study obesity-related diabetes, mainly vascular changes in DR.

Disadvantages: infertility of obese males; males are more prone; female littermates do not develop diabetes; high mortality; monogenic model.

Goto–Kakizaki (GK) rats

The GK model was developed by mating Wistar rats, which were at the upper limit of the normal distribution of glucose tolerance [35,36]. Chronic exposure to a high level of glucose impairs the functions of beta cells and insulin and contributes to the development of hyperglycaemia, which leads to insulin resistance. The pancreatic islets of GK rats can be developed in the form of the so-called “starfish-shaped islets”. They are characterised by a damaged structure, with evident fibres separating the strips of endocrine cells, making them look like starfish [36]. These changes develop with age and are usually invisible in young individuals. In adult GK rats, the number of pancreatic beta cells diminishes, and the insulin reserve in the pancreas decreases by approximately 60% [77]. Research has shown that this is not associated with increased apoptosis of pancreatic beta cells but rather with a decrease in replication of these cells [78]. Secretory changes in the beta cells of GK rats reflect what is happening in the organism exposed to a diabetic environment: reduced differentiation of pancreatic beta cells, chronic exposure to mild hyperglycaemia, and an increased level of nonesterified fatty acids in plasma (i.e., the process defined as glucolipotoxicity).

GK rats are nonobese animals with congenital impaired glucose tolerance. They are a model of mild diabetes and are good model to reflect T2D in humans [33].

An increase in the retinal mean circulation time and a reduction in retinal flow were observed in four-month-old rats [79]. Increased BRB permeability and retinal cell loss were also evident in four-month-old rats after the onset of hyperglycaemia [80,81]. At the age of 30 weeks, lipid and collagen accumulation in the retina and choroid, microgliosis, and upregulation of VEGF were detected in GK rats [82]. The number of retinal vessels increased in animals at six and seven months of age. Moreover, the mRNA expression of HIF-1α, bFGF, platelet-derived growth factor, matrix metalloproteinase 2/9, and insulin–epithelium-derived factor was increased at the same age [83]. At 12 months of hyperglycaemia, a significant reduction of pore density in the RPE cell layer was observed, and it was associated with microglia/macrophage accumulation in the subretinal space, together with the vacuolisation of RPE cells and disorganisation of photoreceptor outer segments [84]. At 8–16 weeks of age, functional abnormalities in electroretinography studies were observed, and the b-wave and oscillatory potential (OP) amplitudes decreased during this period [81].

Suggested application: to study retinal circulation over an extended time.

Disadvantages: various microvascular complications; the exact genetic background is unknown (generated by multiple inbreeding crosses).

Wistar Bonn/Kobori (WBN/Kob) rats

WBN/Kob rats are animal model for spontaneous T2D. At the age of 17 months, only males develop diabetic symptoms, such as hyperglycaemia, glycosuria, hypoinsulinaemia, and glucose intolerance. Around the pancreatic ducts and blood vessels, fibrosis leading to degeneration of the pancreatic islets was found [37]. The WBN/Kob rat is a useful model for both DR and diabetic choroidal angiopathy [85].

In male WBN/Kob rats, newly formed vessels were observed within the retina, and hyalinisation was confirmed in dilated intraretinal vessels after 12 months of hyperglycaemia. Choroidal neovascularisation penetrating through the pigment epithelial layer was observed in male diabetic rats. At 30 weeks of age, retinal degeneration, including the loss of photoreceptor cells and outer nuclear layer cells, was detected [85]. Interestingly, the symptoms of retinal degeneration appeared much earlier than the vascular changes did. All layers of the central and peripheral retina gradually decreased in thickness at 5–45 weeks of age [86]. At around 15 months of age, opacity of the lens began to occur, leading to complete cataracts at 24 months of age [87].

Suggested application: to study neuronal degeneration in DR, mainly photoreceptor apoptosis and diabetic choroidal angiopathy.

Disadvantages: studies limited to males only.

Otsuka Long-Evans Tokushima fatty (OLETF) rats

OLETF rats are a cholecystokinin (CCK)1 receptor knockout model characterised by late-onset hyperglycaemia, polyuria, polydipsia, and mild obesity [38,39]. OLEFT rats spontaneously develop T2D through hyperphagia due to a lack of CCK [38].

Retinal damage was observed at 27 weeks of age in OLETF rats. The total retinal thickness and the retinal nerve fibre layer were significantly thinner at 27 weeks and further decreased in a time-dependent manner. The upregulation of GFAP expression and the elevated rate of apoptosis in the outer nuclear layer were observed in the diabetic retina after 27 weeks of hyperglycaemia [88,89]. Vascular leakage and basement membrane thickening in retinal capillaries were observed after 52 weeks of hyperglycaemia [90]. Moreover, endothelial cell damage, microaneurysms, and loop formations were observed at five months of age [91], and the retinal VEGF mRNA level was 2.2 times higher than that in the control rats [92]. Conversely, the number of acellular capillaries was found to remain morphologically normal, and pericyte ghosts were barely detectable in 45-week-old diabetic rats [93]. A prolonged peak latency of OPs in sucrose-fed OLEF rats was also observed [94].

Suggested application: to study obesity-related diabetes.

Disadvantages: is not suitable for examining angiopathic DR; diabetes develops later in life; inherited mostly in males; monogenic model.

Nonobese spontaneously diabetic Torii (SDT) rats

SDT rats are haracterized by hyperglycaemia and hypoinsulinaemia in the absence of ketonuria without insulin treatment and can serve as a model of nonobese T2D. SDT rats also exhibit glucose intolerance prior to the onset of diabetes [40]. The males are more sensitive, and 100% of individuals develop the characteristic symptoms of diabetes up to 40 weeks of age. Female SDT rats are more resistant, with only 33% of females having symptoms of diabetes at the same age [95]. This model is particularly useful in research on retinopathy. Retinal detachment with fibrous proliferation similar to that occurring in human diabetes was observed [95]. Furthermore, massive haemorrhages in the anterior chamber are typical in SDT rats [82]. The vitreous body was shrunken, and the capillary vessels found in the fibrovascular membrane had thin walls. Dilated retinal vessels and newly formed capillaries were found [95]. Acellular capillaries and pericyte loss were observed in 60-week-old rats [96]. At 44 weeks of age, diabetic animals showed functional changes in the retina, including reduced a-waves, b-waves, and OP amplitudes. Moreover, the implicit times of OPs were prolonged [95,97].

Suggested application: to study proliferative DR.

Disadvantages: gender bias (females are more resistant).

#### 2.1.2. Mouse Models

Type 1 diabetes

Nonobese diabetic (NOD) mice

The NOD mouse model was developed in a laboratory in Osaka in 1974 [41]. In these animals, the inflammation of the islets of Langerhans (insulitis) appears at approximately four weeks of age. Before the diabetic phase, the pancreatic islets are infiltrated by the immune system cells, mainly lymphocytes, NK, and B cells [41]. Inflammation of the islets of Langerhans leads to the destruction of pancreatic beta cells, and diabetes develops violently: mice lose weight quickly and require insulin administration. The destruction of beta cells occurs at approximately 11–14 weeks of age, but full diabetes can develop even at 30 weeks of age [33].

Unlike other autoimmunologic models, this model develops T1 diabetes spontaneously, similar to the way it happens in humans [33]. Both in humans and in the NOD mouse model, the factor responsible for the development of diabetes is the MHC complex, specifically the insulin-dependent region RT1 B/D, which is denoted as iddm1, and its mouse equivalent idd1 [98,99]. Moreover, more than 40 loci have been located in both mouse genes and human genes, and they play an important role in susceptibility to T1D, including the effects of immune system regulation on the pancreatic beta cell function [100]. The similarity between the mouse genome and the human genome is useful in the analysis of the mechanisms of the development of T1D. For this reason, this model is commonly used in research on diabetes therapies based on the immune response [33].

However, this model is not perfect. A number of medicines have been shown to be effective in NOD mice but completely ineffective in humans [101]. Interestingly, when selecting animals for experiments, it is necessary to consider the fact that 80% of females and only 20% of males in this model develop diabetes up to 30 weeks of age [102]. Moreover, in recent years, there have been reports describing the new modification of this model in which proinflammatory cytokines, IL-1β and TNFα, are intravitreally injected into NOD mice [103,104].

Retinal capillary basement membrane thickening and pericyte, endothelial cell, and ganglion cell apoptosis were detected in NOD mice after four weeks of hyperglycaemia, but these retinal changes became more obvious at 12 weeks of age [42]. Constriction of retinal arterioles with a significant reduction in retinal blood flow was observed after three weeks of hyperglycaemia and was mediated by vasoconstrictors, such as thromboxane and/or angiotensin II. Retinal vascular changes were seen in arterioles closely paired with venules [43]. Apart from vasoconstriction, abnormal microvessels were found after four months of hyperglycaemia [105]. After two weeks of hyperglycaemia, Zorrilla-Zubilete et al. found a retinal increase in VEGF expression and loss of the brain-derived neurotrophic factor in NOD mice. Moreover, they found that NAD-dependent sirtuin deacylase (SIRT6) was downregulated [106]. SIRT6 plays a key role in the regulation of glucose metabolism and controls the acetylation levels of histones H3K9 and H3K56. Deficiency of SIRT6 causes retinal transmission defects concomitant to the changes in the expression of glycolytic genes and glutamate receptors, as well as elevated levels of apoptosis in inner retinal cells [107].

Suggested application: to study therapies for diabetes based on the immune response.

Disadvantages: gender bias (males are more resistant); the onset of diabetes is less predictable; the incidence of diabetes depends on conditions of keeping animals (decrease in specific pathogen-free conditions).

Akita mice

The Akita mice were raised as a result of a spontaneous mutation in an allele of the insulin 2 gene. This mutation leads to abnormal processing of the proinsulin peptide, which results in the aggregation of misfolded proteins, followed by endoplasmic reticulum stress. In this model, insulin-dependent diabetes develops at 4–6 weeks of age [44], presenting characteristic symptoms for this type of diabetes: hyperglycaemia, hyperinsulinemia, polydipsia, and polyuria [31].

As the Akita mouse model illustrates the macrovascular changes and neuropathy developed as a result of T1D, it is most frequently used in research on this disease. Nevertheless, due to the low bone weight and impaired healing of fractures, it is an invaluable model for research on bone complications in T1D [44].

This DR model shows signs of both early and late-stage DR. As early signs of DR were observed, vascular permeability, pericyte loss, microaneurysms, acellular capillaries, and neuronal changes also increased. Neovascularisation and new capillary bed formation were observed in the proliferative DR stage of the model [108,109,110]. Retinal ganglion cells (RGCs) were lost from the peripheral retina within the first three months of diabetes, and the dendrites of surviving large ON-α-RGCs underwent morphological changes [111]. The DR features detected in the retinas of Akita mice are similar to those observed in human pathology [110].

Suggested application: to study vascular changes, neuropathy, and bone complications in diabetes.

Disadvantages: gender bias (females are more resistant); the low bone weight and impaired healing of fractures; monogenic model.

Type 2 diabetes

Japanese Kuo Kondo (KK) mice

A model of T2D, the KK mouse strain is characterised by slight obesity (<60 g), and the mutation responsible for the KK phenotype remains unknown [112]. KK mice develop an impaired tolerance of glucose characterised by severe insulin resistance and hyperinsulinaemia [45,46]. The KK mice show signs of compensation of beta cells, with an increased quantity of insulin in the pancreas and Langerhans islet hypertrophy [46].

The best examined substrain is the KK mouse that has the yellow Ay obesity gene. When raised on a high-fat diet, the KK strain shows a considerably increased level of insulin (>1000 µU/mL) [45]. The KK mouse strain is an excellent model for research on the development of obesity-related diabetes.

An increased number of apoptotic cells and capillary basement membrane thickening were observed in the RGC layer in diabetic KK mice after one and three months of hyperglycaemia, respectively [113]. Loss of pericytes, vascular leakage, and retinal acellular vessels were detected after 12 weeks of diabetes [114]. Eight proteins were differentially expressed in the KK-Ay mouse serum, including the serine proteinase inhibitor A3K (SERPINA3), which increases the permeability of retinal microvascular endothelial cells. SERPINA3 is expected to play a significant role in the pathogenesis of diabetes and/or DR [115]. Currently, this model is rarely used in DR studies.

Suggested application: to study obesity-related diabetes.

Disadvantages: hyperphagia and obesity in young are more pronounced in males than in females.

The db/db (Lepr^db^) mice

The db/db mice spontaneously develop type 2 diabetes. These mice are characterized by a mutation of the leptin receptor and develop obesity and diabetes after 4–8 weeks [47]. It has been reported that 16-week-old db/db mice showed pericyte and endothelial cell loss [116] and additionally, increased blood flow in the retina at 18 weeks [117]. Cheung et al. observed the increase in density of retinal capillaries in the inner nuclear layer in 15-month-old mice [118]. Whereas, Yang et al. did not find any obvious microvascular changes in db/db mice alive for 28 weeks, suggesting that the blood retinal barrier was undamaged. He was paying attention to the deficit of RGCs functions visible in the reduction of P1 amplitudes in PERG and the increase in apoptotic RGCs, indicating that the neurodegenerative changes preceded the sever vascular abnormalities [48]. Some researchers reported that a reduction in the number of RGCs and an increased thickness in the central retina are observed even after 6 weeks [119]. These results are in agreement with those reported by Bogdanov et al. [120]. They found a 24% reduction in the number of cell bodies in GCL at 8 weeks, which increased at 16 weeks to 29%. Apart from the loss of cells, they also reported a reduction in amplitude and prolonged implicit time of the b-wave and OPs and a progressive increase of glutamate accumulation. Therefore, they suggest that this model seems to be appropriate for investigating the underlying mechanisms of diabetes-induced retinal neurodegeneration [120]. Moreover, in the db/db mice, clock function and circadian regulation of gene expression is disturbed in the retina [121].

Suggested application: to study diabetes-induced neurodegeneration.

Disadvantages: high mortality (animals do not survive longer than 8–10 months); hormone growth deficiency; hypothermic; infertility.

### 2.2. Selected Chemically Induced Models

Induced models involve the artificial generation of the phenotype that resembles the desirable pathological process. These models are frequently applied in laboratory practice. Artificially induced disease, although significantly different in aetiology from those appearing naturally, allows the examination of the pathological process and implementation of new therapies.

Chemically induced models are widely used in the study of DR. To create a model of this type, animals are usually administered intraperitoneally or intravenously with chemical substances that have a toxic effect on the β-cells of the pancreatic islets. The most commonly used substances are STZ and alloxan. Therefore, in this review, we focused on models that use these chemical inducers for diabetes development. Both chemical compounds are cytotoxic equivalents of glucose that tend to accumulate in pancreatic beta cells through the glucose transporter 2 (GLUT2) [33,49]; Figure 1.

#### 2.2.1. STZ

STZ, a substance discovered in 1956, is an antibiotic synthesised by the Gram-positive bacteria *Streptomycetes achromogenes*. It has a broad spectrum of antibacterial activity [50]. Its strong diabetogenic properties were discovered almost a decade later in 1963 [122].

STZ blocks insulin secretion, causing diabetes [49]. STZ is used to induce both T1D and T2D, depending on the applied dose [33,123]. The intraperitoneal or intravenous administration of 40–60 mg/kg doses causes the insulin-dependent type of diabetes (T1D) in rats or mice, accompanied by changes in insulin secretion and glucose concentration. Animals with STZ-induced diabetes do not require insulin to survive. Many studies have shown that a single STZ dose damages pancreatic cells only partially [124], and diabetes occurs as a result of glucotoxicity (i.e., the state in which damaged β-cells are unable to produce insulin) rather than severe STZ toxicity [51]. The available literature describes a great number of protocols that differ in terms of the applied STZ dose and injection frequency, which determine the rate of development of the disease and its intensity. Note that STZ sensitivity in rodents is different. Rats develop full symptomatic diabetes much faster than mice do. Moreover, the STZ sensitivity of the Langerhans islets differs depending on the mouse strain. The resistance of female mice to STZ is higher than that of male mice [125]. Therefore, it is crucial to select an appropriate protocol for the type of research that we want to perform. The most common protocols are listed in the Table 3.

STZ is a combination of the highly reactive nitramine methylnitrosourea moiety in the C2 position and the glucose molecule. Nitramine methylnitrosourea is responsible for the toxic properties of STZ, while the glucose molecule leads the substance straight to the pancreatic β-cells [121]. STZ specifically targets pancreatic β-cells, and it recognises GLUT2, which appears abundantly in the plasma membrane of β-cells [33,49,50]. As GLUT2 is also present in the liver and kidneys, high STZ doses are expected to impair their function; however, they are quickly metabolized and excreted by the kidneys [50]. The inhibition of the expression of the GLUT2 transporter in the cell membrane protects pancreatic β-cells from the cytotoxic and diabetogenic activities of STZ [157].

After STZ enters the pancreatic β-cells, it breaks the continuity of the DNA thread, which stimulates the nuclear poly (ADP-ribose) synthetase and lowers the level of NAD^+^ and NADP^+^, which are nucleotides that perform key roles in cellular respiration processes; Figure 2. This plays a significant part in the inhibition of proinsulin synthesis and induces diabetes. As a result of the decrease in NAD^+^ and NADP^+^ levels, oxygen is activated in cells, and hydrogen peroxide is generated [51]. Reactive oxygen species (ROS) may also form in cells during STZ accumulation in β-cells because of their chemical instability [158,159]. According to research, the positions of N^7^ nitrogen and O^6^ oxygen of guanine are especially exposed to STZ alkylating molecules [159,160].

STZ diabetes can be reversed by applying insulin therapy [50]. Research on a mouse model of diabetes with STZ use has shown that glycaemic control through insulin can contribute to the regeneration of pancreatic β-cells. The dysfunction of pancreatic cells results from hyperglycaemia, and when it is reversed by insulin administration, the functions of β-cells improve and they are regenerated [161].

In STZ-induced diabetic mice, the optical coherence tomography image analysis at six weeks after DM induction showed thinning of the nerve fibre layer (NFL) and GCL, but immunohistochemistry showed no reduction of RGC density. However, 20 weeks after DM induction, the thinning of NFL and GCL was even greater, and a significant loss of RGC density was observed. Moreover, microvascular changes occurred 18–24 weeks after DM induction [162]. Yang et al. observed progressive loss of RGC at 6–12 weeks after the induction of diabetes [163]. However, some studies did not find evidence of RGC death due to prolonged diabetes [164,165]. GFAP upregulation in glial cells and reactive gliosis occurred at 4–5 weeks of hyperglycaemia [166]. Four weeks after DM induction, functional abnormalities in electroretinography studies were observed, including decreased OP3, prolonged implicit time of OP2 and OP3 [167], and decreased a-waves and b-waves [168]. Numerous leukocytes, leukocyte clumps, and pericyte loss were found in diabetic retinal capillaries at four weeks of hyperglycaemia [167]. After six weeks, acellular capillaries were present, and basement membrane thickening of the retinal vessels increased in long-term observation [169,170,171]. In the rat STZ model, similar morphological and functional changes were observed. Interestingly, rats were more sensitive to STZ; therefore, all DR signs were seen earlier than they were in the mouse model [137,138,139]. Gastinger et al. reported that after only two weeks, the retinas of STZ diabetic rats had 2.5-fold more TUNEL-positive nuclei compared to the control and that dopaminergic and cholinergic amacrine cells were lost during the early stages of DR [172]. Moreover, after three months of hyperglycaemia, RGC loss was associated with morphological changes. The surviving RGC showed significant dendritic field enlargement as a compensatory response to the overall loss of cells in diabetes [140]. The amplitude of the a-wave decreased from 31% at four weeks of diabetes to 53% in 36 weeks, and that of the b-wave to 50% [139]. After one week of hyperglycaemia, there was a significant reduction in the RGC positive scotopic threshold response [52]. After two weeks, the OPs were reduced [173].

Suggested application: to study the early stage of DR; the most widely used model in DR research.

Disadvantages: hyperglycaemia, rather than insulin resistance, develops as cytotoxic action on β-cells; induced diabetes is less stable and reversible by spontaneous regeneration of β-cells; gender bias (female mice are more resistance).

#### 2.2.2. Alloxan

Similar to STZ, alloxan is an organic glucose analogue and urea derivative. Moreover, it is carcinogenic and cytotoxic [49]. Alloxan was isolated for the first time in 1818 and described in 1838 [174]. Owing to its structural similarity to the glucose molecule, it is transported through the diffusive transport mechanism, with the participation of a transport protein—GLUT2. Interestingly, alloxan does not have any influence on the impairment of GLUT2 activity and shows an ability to be selectively uptaken by pancreatic β-cells, which significantly enhances the uptake by β-cells, leading to accumulation in these cells [174,175,176,177]. Due to its attractive price and high availability, alloxan was one of the substances most commonly used to create chemically induced models.

Diabetes induced by alloxan is an insulin-dependent variant (T1D), and its preparation has been tested on many animal species, including rodents, dogs, and monkeys [53,178]. The model induced by alloxan can be obtained through the injection of a single dose or repeated administration—intraperitoneally, intravenously, or subcutaneously. However, the most frequently chosen option is intraperitoneal administration (Table 3). The doses administered in individual studies fluctuate at 90–200 mg/kg of animal body weight (BW) [52], but many studies have reported doses of 170–200 mg/kg BW as the most effective in a single application [179].

The mechanism of alloxan action is based on the partial degradation of β-cells in the Langerhans islets. Two pathological effects work here. The first effect is associated with the selective impairment of glucokinase, a glucose sensor enzyme, by alloxan, which results in the impairment of insulin secretion. The second effect is based on the production of ROS, which promotes cell death [180,181]. Alloxan is a very unstable compound that undergoes redox cycles. In the presence of thiols (i.e., organic sulphur compounds, alcohol analogues, and glutathione (GSH)), alloxan undergoes a cyclic reaction that generates ROS, such as superoxide anion radicals or hydroxyl radicals. The reaction involves the reduction of alloxan to dialuric acid and the re-oxidation of dialuric acid to alloxan [161]. Alloxan accumulation and, consequently, the reactive forms of oxygen in pancreatic β-cells result in their death by necrosis; Figure 3. However, death does not result from the alloxan activity itself but rather from the glucotoxicity of β-cells caused by the redox cycle and toxic ROS [49].

Nevertheless, a certain paradox has been observed. In rodent research, thiols, including cysteine and GSH, are applied to prevent the development of diabetes induced by alloxan [182,183]. When the concentration of reducing substances in the blood or in the extracellular space is significantly increased through thiol administration, more of the alloxan is reduced extracellularly; thus, a smaller amount of preparation accumulates in cells [49]. Protection against the toxic effects of alloxan is also provided by glucose, as confirmed by in vitro and in vivo studies [184]. It occurs because of the prevention of glucokinase impairment, which results in the preservation of antioxidative mechanisms of β-cells [176,177].

Currently, alloxan is rarely used to induce the DM model because of its many inconsistencies and anomalies [174,185]. After three months of hyperglycaemia induced by alloxan, no apoptosis of RGC and no changes in blood vessel permeability were detected in mice, but a decrease in the length of microglia cell processes was observed. However, functional changes in the retina were observed, such as a reduction in OPs and the b/a amplitude wave ratio [186]. Moreover, retinal tissues showed an altered expression of inflammatory, oxidative, and proliferative markers (i.e., TNF*α*, IL6, NF-κB, iNOS, and Tp53) [149]. In the alloxan-induced rat model, after three months, dilated and congested blood vessels, oedema, destructive damage, and formation of papillary structures were observed in the retinal GCL [151]. A reduction of the number of blood vessels of the choroid, with pathological alterations of the endothelial cells and vascular walls, was observed in the early stages of the disease (i.e., 30 days), and this did not change as time progressed [156]. Pericyte loss, acellular capillaries, and membrane thickening were detected after 12 months of hyperglycaemia [187].

Suggested application: to study the early stage of DR;

Disadvantages: cytotoxic effects on other body organs; hyperglycaemia, rather than insulin resistance, develops as cytotoxic action on β-cells; induced diabetes is less stable.

## 3. Conclusions

DR is a common diabetes complication that considerably affects patients’ quality of life and generates huge costs associated with the treatment. The costs are not limited to the provision of ophthalmological care but also to psychological or psychiatric care. Understanding the mechanisms leading to the development of vascular and neuronal changes in the retina accompanying diabetes is crucial to delay the progression of the disease. The animal models used in the study of DR are helpful tools that allow researchers to analyze the mechanisms underlying the disease and test the efficiency of new therapeutic strategies. This article describes the rodent models most commonly applied in laboratory practice. Chemically induced models, mainly those using STZ and alloxan, are often used in scientific studies. This review examines a collection of protocols available in the literature, with detailed descriptions of the amounts of administered doses and the frequencies of administration of these chemical substances. Aside from rodents (mice and rats), other animal species are also used as model animals. The available literature includes protocols designed for rabbits [188,189], dogs [190,191], cats [192], pigs [193,194], and monkeys [195,196]. However, researchers usually choose mouse or rat models. The choice is primarily based on the low breeding costs. As most of the research is carried out on rodent models, comparative analyses of the results obtained by different research teams can be performed. However, even models based on the same species show a huge variability depending on the mechanism of disease induction. Moreover, within the same species, significant differences are also observed depending on the strain, age, and sex of the species selected for testing. Despite these inconveniences, animal models are still an important and irreplaceable research tool. Note that the choice of a research model should be thoroughly considered and dependent on the aspect of the disease to be analyzed.

## Figures and Tables

**Figure 1 biology-12-00262-f001:**
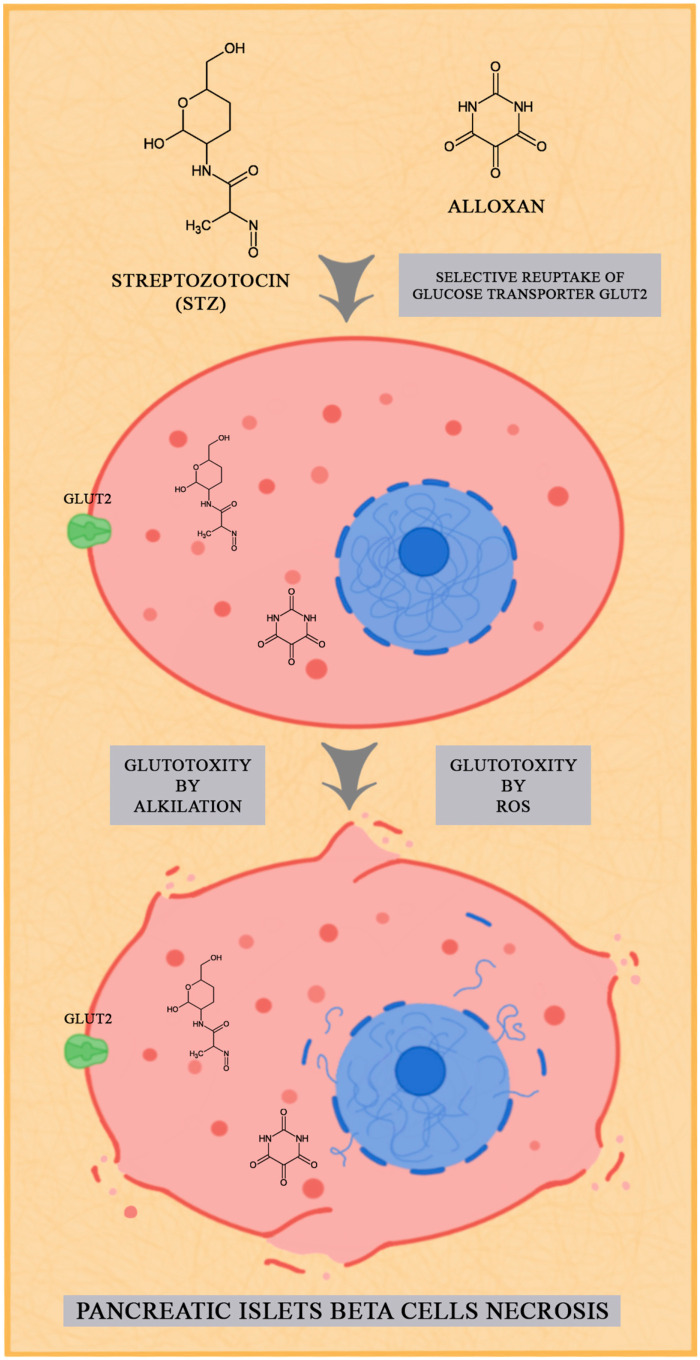
The effect of streptozotocin and alloxan on pancreatic islets beta cells.

**Figure 2 biology-12-00262-f002:**
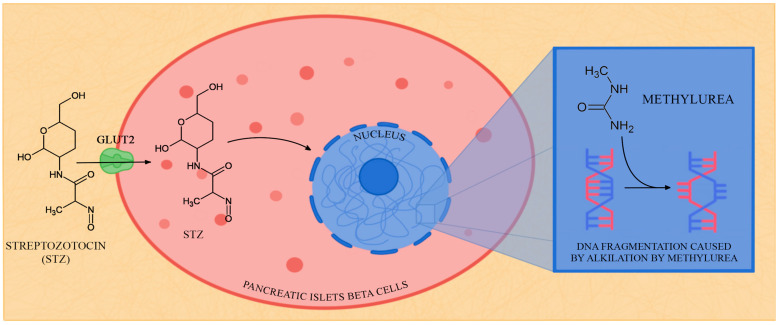
The mechanism of streptozotocin action in pancreatic islets beta cells.

**Figure 3 biology-12-00262-f003:**
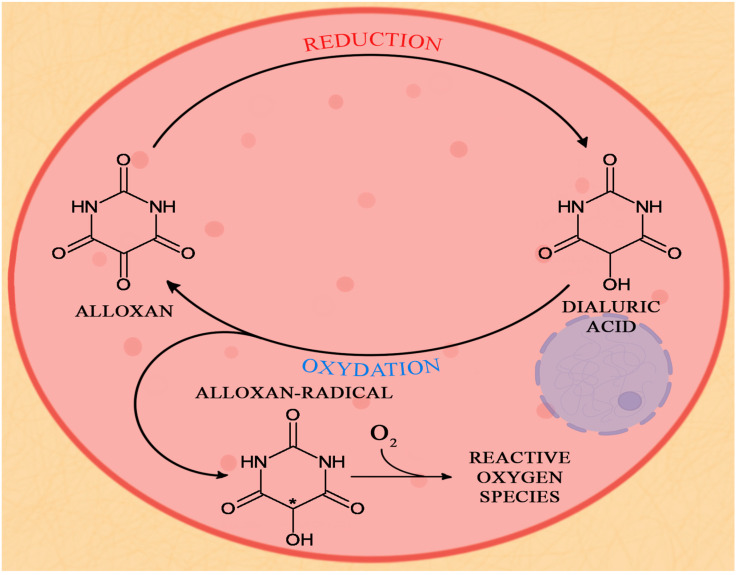
The redox cycle of alloxan in pancreatic islets beta cells.

**Table 1 biology-12-00262-t001:** Criteria for diagnosing the stage of development of diabetic retinopathy.

Classification of DR	Features
No retinopathy	no microvascular lesion
Mild nonproliferative diabetic retinopathy (NPDR)	microaneurysms only
Moderate NPDR	Microaneurysms retinal haemorrhages hard exudates
Severe NPDR	more than 20 haemorrhages in each of the four quadrants, intraretinal microvascular abnormalities (IRMA) in one quadrant venous and arterial changes venous beading in two quadrants
Proliferative diabetic retinopathy (PDR)	neovascularization of optic disc or retina preretinal haemorrhage vitreous haemorrhage

**Table 2 biology-12-00262-t002:** The main types and features of rodent models of diabetic retinopathy.

Model Type Depending on the Induction Mechanism	Model	Type of Diabetes Mellitus	Features	References
Genetic Models	Rats Models			
	Biobreeding (BB) rats	1	the diabetes-prone (BBDP) rats —diabetes develops spontaneously; the diabetes-resistant (BBDR) rats—trigger required to develop diabetes; lymphopenia, hyperglycaemia, hypoinsulinemia, ketonuria, weight loss	[31,32]
	Zucker diabetic fatty (ZDF) rats	2	leptin receptor mutation hyperphagia, obesity, hyperinsulinemia, hyperlipidaemia	[33,34]
	Goto–Kakizaki (GK) rats	2	congenital impaired glucose tolerance, hyperglycaemia, insulin resistance, starfish-shaped isles	[35,36]
	Wistar Bonn/Kobori (WBN/Kob) rats	2	diabetes develops spontaneously, only males develop diabetes hyperglycaemia, glycosuria, hipoinsulinaemia and glucose intolerance; fibrosis of the pancreatic ducts and blood vessels → degeneration of the pancreatic islets	[37]
	Otsuka Long-Evans Tokushima fatty (OLETF) rats	2	a cholecystokinin (CCK)1 receptor knockout model, late onset hyperglycaemia, polyuria, polydipsia and mild obesity	[38,39]
	Nonobese spontaneously diabetic Torii (SDT) rats	2	males are more sensitive, hyperglycaemia and hipoinsulinaemia	[40]
	Mice Models			
	Nonobese diabetic (NOD) mice	1	diabetes develops spontaneously, females are more sensitive, Langerhans islets inflammation; intraperitoneal injection of IL-1β and TNFα, stimulates the development of this model	[41,42,43]
	Akita mice	1	spontaneous mutation in an allele of the insulin 2 gene (abnormal processing of the proinsulin peptide), hyperglycaemia, hyperinsulinemia, polydipsia and the polyuria	[31,44]
	The Japanese Kuo Kondo (KK) mice	2	severe insulin resistance and hyperinsulinemia, slight obesity, Langerhans islet hypertrophy	[45,46]
	The db/db (Lepr^db^) mice	2	mutation of the leptin receptor obesity, hyperglycaemia, atrophy of pancreatic β-cells, hypothermic, hormone growth deficiency	[47,48]
Chemically Induced Models	Streptozotocin (STZ): High single dose STZ Multiple low-dose STZ	1 and 2	STZ blocks insulin secretion (glucotoxicity, DNA alkylation, ROS production), rats are more sensitive than mice are, female mice are more resistant than males are. STZ diabetes can be reversed by applying insulin therapy	[49,50,51]
	Alloxan	1 and 2	carcinogenic and cytotoxic substance partial degradation of β-cells in Langerhans islets’ selective impairment of glucokinase and ROS production	[52,53]

**Table 3 biology-12-00262-t003:** The current protocols for streptozotocin and alloxan used in the DR research.

Chemically Induced Model	Dose	Way of Application	Strain	Level of Hyperglycaemia	Reference
Streptozotocin	Mice				
	10 mg/kg BW ^1^ for 5 days	IP ^2^	BALB/c or C57BL/6 mice—male (4–5 weeks old)	>200 mg/dL * 438 ± 70 mg/dL (fed with high-fat diet) after 6 weeks; >600 mg/dL (fed with low-fat diet) after 6 weeks ^†^	[17] ^†^, [126] *
	55 mg/kg BW for 5 days 75 mg/kg BW for 5 days	IP	C57BL/6J mice (10–12 weeks old) male female	547 ± 65.9 mg/dL (male) 237 ± 86.3 mg/dL (female)	[125]
	60 mg/kg/BW for 5 days	IP	SNS-HIF1α^−/−^ or HIF1α^fl/fl^ mice (7–9 weeks old) C57BL6/j mice—male (7–8 weeks old)	380–500 mg/dL after 2 weeks * 300–400 mg/dL after 4 weeks ^†^	[127] * [128] * [129] ^†^
	85 mg/kg BW for 3 days	IP	hy1-YFP-H transgenic mice (maintained under the C57BL/6J background)—male and female (6–7 weeks old)	343 ± 14.04 mg/dL after 4 weeks	[130]
	single dose—150 mg/kg BW	IP	C57BL/6 mice—male (6–8 weeks old) BALB/c mice—male (6–8 weeks old)	26.61 ± 1.14 mmol/L → 478.98 ± 25.2 mg/dL * ≥16.7 mmol/L → 300.6 mg/dL ^†^	[131] *, [132] ^†^
	single dose—200 mg/kg BW	IP	C57BL6 mice (8–12 weeks old)—male	28.9 mmol/L → 520.2 mg/dL	[133]
	Rats				
	single dose—30 mg/kg BW	IP	Sprague-Dawley rats—male (6 weeks old)	20.39 ± 1.72 mmol/L → 367 ± 31 mg/dL after 1 week	[134]
	single dose or double dose—50–65 mg/kg BW	IV ^3^	Sprague-Dawley rats—male (6–23 weeks old)	50 mg/kg (double dose) → 428 ± 18 mg/dL (6–11 weeks old) 65 mg/kg (single dose) → 465 ± 26 mg/dL (6–11 weeks old) or 464 ± 23 mg/dL (12–17 weeks old) or 444 ± 6 mg/dL (18–23 weeks old)	[135]
	single dose—60 mg/kg BW	IP	Sprague-Dawley rats (2–3 months old)	25.2 ± 3.1 mmol/L → 453.6 ± 55.8 mg/dL after 4 weeks *; 265 ± 12 mg/dL after 4 weeks ^†^ 465 ± 17 mg/dL after 3 days ^‡^	[136] *, [137] ^†^, [138] ^‡^, [139] *
	single dose—65 mg/kg BW	IP	Sprague-Dawley rats—male	23.2 ± 0.7 mmol/L → 417.6 ± 12.6 mg/dL after 12 weeks	[140]
	single dose—90 mg/kg BW	IP	Sprague-Dawley rats—male	38.9 ± 2.1 mmol/L → 700 ± 37.8 mg/dL after 20 weeks	[141]
	45 mg/kg BW 55 mg/kg BW	IP	Wistar rats—male (2 months old)	25.6 ± 8.2 mmol/L → 460 ± 147.6 30.1 ± 7.7 mmol/L → 541.8 ± 138.6 after 4 weeks	[142]
	single dose—55 mg/kg BW	IP	Wistar rats (8 weeks old)	25.1 ± 3.1 mmol/L → 451.8 ± 55.8 after 4 weeks	[139]
	single dose—60 mg/kg BW	IP	Wistar rats (10–12 weeks old)	300 mg/dL after 10 days	[143]
	single dose—60 mg/kg BW	IV	Wistar rats—male (8 weeks old)	426 ± 31 mg/dL	[144]
Alloxan	Mice				
	single dose—150 mg/kg BW	IP	BALB/c mice—female (6–8 weeks old)	>250 mg/dL after 1 day	[136]
	single dose—180 mg/kg BW	IP	BALB/c mice—male (6–8 weeks old)	>220 mg/dL after 2 days	[145]
	single dose—186.9 mg/kg BW	IP	BALB/c mice—male (6–8 weeks old)	>200 mg/dL after 12 h >260 mg/dL after 1 day	[146]
	single dose—50 mg/kg BW	IV	ICR mice (known as Swiss CD-1 mice)—male (5 weeks old)	>500 mg/dL after 6 weeks	[147]
	single dose—150 mg/kg BW	IP	ICR mice—male (4 weeks old)	465.45 ± 15.60 mg/dL after 2 weeks	[148]
	single dose—180 mg/kg BW	IP	Swiss albino mice—male	≥250 mg/dL	[149]
	single dose—200 mg/kg BW	IP	Swiss albino mice (8 weeks old)	262 ± 4.80 mg/dL	[150]
	Rats				
	120 mg/kg BW for 3 days	IP	Wistar rats—male	290 ± 5 mg/dL * 284.83 ± 6.96 ^†^ after 3 days	[151] *, [152] ^†^
	single dose—150 mg/kg BW	IP	Wistar rats—male	212 ± 2.41 mg/dL after 4 days; 274 ± 2.82 mg/dL after 1 week; 418 ± 3.53 mg/dL after 2 weeks; 465 ± 3 mg/dL after 4 weeks * >500 mg/dL after1 week ^†^	[153] *, [154] ^†^
	single dose—180 mg/kg BW	IP	Wistar rats—male	-	[155]
	single dose—300 mg/kg BW	IP	Wistar rats (16 weeks old)—male	25.98 ± 1.84 mmol/L → 467.84 ± 33.12 mg/dL after 4 weeks; 32.60 ± 0.80 mmol/L → 586.68 ± 14.4 mg/dL after 8 weeks	[156]

^1^—body weight; ^2^—intraperitoneal injection; ^3^—intravenously injection; * ^† ‡^—references describing the indicated level of hyperglycaemia.

## Data Availability

Data sharing not applicable.

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
