# Peer review of "Rodent Models of Diabetic Retinopathy as a Useful Research Tool to Study Neurovascular Cross-Talk"

_biology, 2023, doi:10.3390/biology12020262_

Round 1
Reviewer 1 Report
In this manuscript, the authors extensively reviewed the rodent models of diabetic retinopathy. The manuscript is well written and tables and cartoons have been provided to summarize the results of their peer review.
Only a few comments and suggestions.
In Introduction, for the international classification of diabetic retinopathy, the authors should refer to the International Council of Ophthalmology. ICO guidelines for diabetic eyecare. http://www.icoph.org/downloads/ICOGuidelinesforDiabeticEyeCare, which is largely used both in clinical and scientific settings. Please substitute your current reference 7 with this one.
The authors pointed out that animal models are undoubtedly a useful tool for understanding the mechanisms leading to the development of DR and that “Rodents have many similarities to humans, especially at the metabolic level.” However, the authors should also consider and declare that the commonly used rodent models in research, mouse and rat, do not possess a macula. Therefore, these models may not be useful in understanding the macular complications secondary to diabetes, like the diabetic macular edema.
Reviewer 2 Report
Comprehensive review about the current rodel models of diabetic retinopathy and the dysfunction in the neurovascular unit on them. The information provided is exhaustive and the review very well written and organised. Although several review had addressed the same topic, the authors manage to make the present work different from other published in 2022 and 2017, to name a few.
However, perhaps I miss some extra information on the more used animal models of diabetic retinopathy, as extensive research and characterization has been done in some of them and few information have been provided. For example, the information related to STZ is way more extensive than db/db or Akita model.
Is there any reason not to include high fat diet mouse model?
